



# 1 Exploiting LSPIV to assess debris flow velocities in the field

Joshua I. Theule[1,3], Stefano Crema[2], Lorenzo Marchi[2], Marco Cavalli[2], Francesco Comiti[1]

[1]Faculty of Science and Technology, Free University of Bozen-Bolzano, Bozen-Bolzano, 39100, Italy
[2]Research Institute for Geo-hydrological Protection, National Research Council of Italy, Padova, 35127, Italy
[3]TerrAlp Consulting, 100 chemin du grand pré, 38410 St Martin d'Uriage, France

*Correspondence to*: Joshua I. Theule (joshua.theule@terralpconsulting.com)

**Abstract.** The assessment of flow velocity has a central role in quantitative analysis of debris flows, both for the
characterization of the phenomenology of these processes, and for the assessment of related hazards. Large scale particle
image velocimetry (LSPIV) can contribute to the assessment of surface velocity of debris flows, provided that the specific
features of these processes (e.g. fast stage variations and particles up to boulder size on the flow surface) are taken into
account. Three debris flow events, each of them consisting of several surges featuring different sediment concentration, flow
stage and velocity, have been analyzed at the inlet of a sediment trap in a stream of the eastern Italian Alps (Gadria Creek).
Free softwares have been employed for preliminary treatment (ortho-rectification and format conversion) of video-recorded
images as well as for LSPIV application. Results show that LSPIV velocities are consistent with manual measurements on
the ortho-rectified imagery and with front velocity measured from the hydrographs in a channel reach approximately 70 m
upstream of the sediment trap. Horizontal turbulence, computed as the standard deviation of the flow directions at a given
cross-section for a given surge, proved to be correlated with surface velocity and with visually estimated sediment
concentration. The study demonstrates the effectiveness of LSPIV in the assessment of surface velocity of debris flows, and
permit to identify the most crucial aspects for improving the accuracy of debris flows velocity measurements.

## 1 Introduction

Debris flows are a rapid flow of saturated non-plastic debris in a steep channel (Hungr et al., 2001). They consist of poorly
sorted sediments mixed with water and organic debris with sediment concentrations higher than 50% by volume or 70% by
mass (Costa, 1984; Phillips and Davies, 1991) and can travel over long distances at relatively high velocities (generally
between 2 to 20 m s$^{-1}$) (Iverson, 1997; Rickenmann, 1999). Debris flows are relatively infrequent and complex events which
make it difficult to characterize their dynamic heights, velocities, discharge, and flow resistance of the material, among other
aspects.
Debris-flow velocities and discharge are typically backcalculated from surveyed channel bends with superelevated flow
heights using the forced vortex equation (eg. Hungr et al., 1984; Chen, 1987; Prochaska et al., 2008; Scheidl et al., 2014).
The measured parameters (flow heights, velocity, and slope) from post-event surveys for this equation can also be used to
estimate flow resistance coefficients to understand the viscosity and sediment concentrations of the debris-flows (eg.



Rickenmann, 1999). However, sediment concentrations are known to significantly increase and decrease during the propagation of the flow (eg. Pierson and Scott, 1985; Rickenmann et al., 2003) and the velocity profile of the surges can also vary which limit the reliability of post-event field methods.

High-frequency monitoring projects of debris flows are growing because of their increasing feasibility and capability for observing several parameters of this complex process (eg. Marchi et al., 2002; Arattano et al., 2012; Navratil et al., 2013; Comiti et al., 2014). Typical monitoring stations consist of geophones, ultrasonic sensors (or radar), and video cameras which satisfy the basic measurements of velocity, height, discharge, and visual validation. Some catchments even have multiple stations distributed throughout the debris-flow channel and some located in headwater channels (Berti et al., 2000; Marchi et al., 2002; Hürlimann et al., 2003; McCoy et al., 2010; Arattano et al., 2012; Navratil et al., 2013; Comiti et al., 2014).

Video imagery originally started as a validation of the instrumented recordings and visual interpretation, but as cameras, power, and storage capacities improve, further analysis can be made. Manual tracking of particles with field measurements can measure velocities accurately when compared to stage sensors (eg. Arattano and Grattoni, 2000; Marchi et al., 2002). The video imagery of debris flow can also be used to interpret the turbulence, sediment mixture, sediment concentration, presence of rigid plugs and laminar flows (eg. Marchi et al., 2002). Horizontal velocity distributions from video imagery have shown variations of flow resistance between events and within the same surge (Genevois et al., 2001). Rheological parameters are known to significantly vary within the same surge, but they are very difficult to quantify in the field.

Large scale particle image velocimetry (LSPIV) is another video imagery technique often used in rivers to measure two dimensional velocities from high resolution images at high frame rates (eg. Fujita et al., 1998; Hauet et al., 2008; Le Coz et al., 2010; Muste et al., 2014). Cross-correlations are made between time-step imagery within a given search window. This is typically applied in steady flows by tracking bubbles, ice, debris, and artificial seeding. Discharge rates can then be estimated because of the stable cross-sections during the flow. LSPIV and series of elevation models were also compared during bedload transport flume experiments to quantify discharge and deposition, as well as Froude and Shield's numbers (Piton, 2016).

These types of analysis are difficult for debris flows because the varying surges can vary in height and significantly modify the channel bed. The LSPIV method was tested on a pulsing flash flood in a stable reach from a GoPro recording that was available on Youtube (Le Boursicaud et al., 2016). There was a 3-5% velocity error for 15-30 cm water level bias which was the largest source of error in the analysis. Recently, a long-term discharge monitoring project of a mountain stream with LSPIV applications used an automated detection of the water level heights (Stumpf et al., 2017). This method still poses a problem for the highly irregular debris-flow surfaces, however considering the low percent error, approximate heights should be feasible for surface velocity.

To our knowledge, the application of LSPIV on debris flows has not been deeply investigated whereas it could provide direct measurement to quantify rheological behavior of debris flows. Our objective is to test the LSPIV method on debris flows using available monitoring cameras in a monitored catchment in the Italian Alps (Gadria catchment) (Fig. 1). The aims of





this work are to explore: 1) the spatial and temporal variation within one study reach of debris-flow surges occurred in the period 2013-2015, 2) a detailed analysis of an individual surge dynamic, 3) the quantification of a "horizontal turbulence index" (influenced by rheological parameters) from the directional variation of vectors, and 4) the limitations/perspectives of the LSPIV for further development.

## 2 Setting

The Gadria catchment is situated in Vinschgau-Venosta Valley (South Tyrol) in the Eastern Italian Alps (Fig. 1A), and features a drainage area of 6.3 km$^2$, with an average slope of 79.1 % (between 1394 and 2945 m a.s.l.). The source area consists of highly deformed and fractured metamorphic rock, thick glacio-fluvial deposits and steep topography which makes the catchment prone to rockfall, landslides, avalanches and debris flows. The topographic settings of the catchment ensure an effective connectivity of sediment between the source areas (D'Agostino and Bertoldi, 2014) and the downstream channel reaches (Cavalli et al., 2013). Debris flows occur in the summer and are usually triggered by spatially-limited convective storms. The mean volume of the debris flows observed between 1979 and 2013 is 14,000 m$^3$ (median 8000 m$^3$) (Aigner et al., 2015). The sediment yield of the Gadria catchment between 2005 and 2011, a period normal as to frequency and magnitude of debris flows, was computed through DEM differencing (Cavalli et al., 2016) and amounted to about 5200 m$^3$km$^{-2}$yr$^{-1}$. Instrumented monitoring of the Gadria catchment began in 2011, refer to Comiti et al. (2014) for detailed information of the study site and monitoring setup.

Two cameras are in a sediment trap (retention basin) near the alluvial fan apex, one looking upstream (Cam1) and the other looking down at a more perpendicular angle to the flow (Cam2). The third camera (Cam3) is in the next reach upstream from the sediment trap at a closer proximity to the flow (Fig. 2). These three cameras are connected to a cabin equipped with power supply and a server (8 Tb storage capacity) collecting all the monitoring data. The fourth camera is in an upstream ravine and it is triggered by a rain gauge when there is at least one minute of rainfall. For this study, we focused on the application of LSPIV using only one of the four MOBOTIX M12 video cameras, Cam 2.

We attempted to utilize also the other cameras for LSPIV application, but Cam 1 and Cam 3 were too low with an upstream view, and this creates very skewed images with little ground coverage for LSPIV. Cam 2 was the best option because it was located higher on top of the levee with an angle more perpendicular to the flow path. Cam 4 was problematic due to the unchannelized nature of the recorded events, coupled to the relative long distance between the camera and the moving sediment.



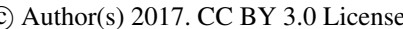



**Figure 1: Gadria catchment is situated in the Vinschgau-Venosta Valley, South Tyrol, Italian Alps (A). The catchment is instrumented with rain gauges, radar stage sensors, video cameras, and geophones (B). At the end of the catchment (C) is a sediment retention basin where most of the instrumentation exists connected to a cabin with a power supply, internet connection, and a server with 8 Tb of storage capacity.**



## 3 Methods

The LSPIV methods that we used are initially based from Le Boursicaud et al. (2016). The previous study tested the LSPIV method on a pulsating flashflood in the French Alps recorded from a GoPro. The videos were treated for photo stitching and format conversion using freeware and the LSPIV calculation on the freeware Fudaa-LSPIV (Le Coz et al., 2014) (https://forge.irstea.fr/projects/fudaa-lspiv/files).

### 3.1 Video treatment

The Mobotix security camera that we used is an IP camera, therefore the frame per second cannot be fixed and they are automatically adjusted to the available light. This initially posed as a problem since our aim was to have a constant 10 frame per seconds (fps). During recording of the flow events, the frequency reduced to 2 - 3 fps because of the low lighting of the storms. We needed a standard frame rate for LSPIV calculations, therefore we subsampled the images to the minimum frame rate of each flow event (Table 1).

Also, the camera had a fisheye lens, therefore significant distortion correction was required. A checkerboard pattern image from the camera was used in a free software Hugin (http://hugin.sourceforge.net) which has a tool for distortion correction. This was then applied to all the video imagery and converted to an ASCII grey scale format using batch processing in the XNview freeware (www.xnview.com). This used to be necessary for the Fudaa software, however it now can handle jpeg and tiff colored formats.

### 3.2 Reference points using Structure from Motion Photogrammetry

High resolution colored point clouds from Structure from Motion (SfM) surveys were found to be very useful for matching reference points with the video images (Fig. 2A). In active debris-flow channels, permanent points are difficult to keep within the active area, and with oblique angled cameras, there needs to be as many reference points as possible. The sediment trap and channel were surveyed before and after flow events by walking up and down the banks with a camera mounted on a 5-m pole with georeferenced targets distributed throughout the channel and trap. The SfM photogrammetry using AgiSoft® Photoscan (eg. Westoby et al. 2012; Javernick et al., 2014; Piermattei et al. 2015) was used to generate high resolution colored point clouds (1300-2900 pts/m$^3$) making it a reliable spatial and visual reference. For the LSPIV purposes, the point clouds were rotated to make a horizontal flow plane. These flow planes are easily visible in the colored point clouds with the distinct mudlines.



**Table 1: LSPIV parameters used for the 2013, 2014, 2015 events.**

|  | 2013 | 2014 | 2015 |
|---|---|---|---|
| resolution | 5cm/pixel | | |
| error near the flow plane | 3-10 cm | 4-7 cm | 8-13 cm |
| # reference points | 13 | 13 | 14 |
| interrogation area | 26 pixel (1.3 m) | | |
| search area (pixels) | 75-100 down; 5 up; 35-50 left; 30-50 right | | |
| time step | 0.333 s | 0.5 s | 0.5 s |
| grid | 0.4-1.2 m | | |
| area | 28-35 m long and 7-32 m wide | | |

## 3.3 Fudaa LSPIV

Targets and natural features were used as reference points for matching between the SfM point cloud (both pre-event and post-event) and video imagery (Fig. 2A, 2B). Corners of rocks next to the flow line were typically used on each side of the channel, and sometimes exposed stable rocks within the channel. Errors increase going down and across the channel according to the camera's oblique angle. The flow plane elevation was also measured by averaging matched features touching the flow line in the post-event point cloud. The unsteady flows required separating the fronts and tails to redefine the flow plane elevation which is known to be the largest source of error for LSPIV (Le Boursicaud et al., 2016).

The interrogation area (IA) is the boundary for calculating a correlation coefficient which needs to be representative of the flow velocity (Fig. 2C). It should find the travel distance of general features in the flow between each time step, not individual particles which is unrealistic in irregular flows with sediment rolling and continuously being submerged. We used a 26 x 26 pixel (1.3 m x 1.3 m) interrogation area for calculating the correlation coefficient and a search area of 75-100 pixel (3.75 - 5 m) downstream, 60 - 100 pixel (3 - 5 m) wide, and a small 5 pixel segment upstream to capture any perpendicular flow.

To have a good spatial distribution of the flow with a manageable dataset, we selected a grid with an approximate spacing of 0.7 m (varies with flow width) (Fig. 2C). Within the Fudaa software, we filtered any velocities with a correlation coefficient less than 0.5-0.6 for a robust dataset (Fig. 2D). The velocity vectors were transferred into ArcGIS and overlaid on the corresponding orthorectified image for manual cleaning. Noisy data can occur outside of the flow area because of rain, wind, changing light reflection on wetted surfaces. The manual treatment of the vectors was also necessary for outlining and separating the different surges and parts of the surge (front and tail) traveling through the study reach.

The spatial distribution of velocity vectors covering the reach provided an opportunity to examine the directional variation of the velocity vectors to characterize the turbulence of the various debris-flow surges (Costa 1984). Since our LSPIV method is in the two dimension, we define it as the horizontal turbulence index ($T_h$). We measure $T_h$ by taking the standard deviation of the flow directions at a given cross-section for a given surge. The given cross-section can be used to examine the changing characteristics of the surges rather than the spatial variation. Therefore, small $T_h$ should represent laminar flows and high $T_h$ are more turbulent flows.



The LSPIV results were taken from cross-section XS (Fig. 1C) to have accurate comparisons of debris flow surges. This is

the most stable cross-section before the widening in the sediment trap. It is also the closest and most perpendicular view

from the camera resulting in the most accurate LSPIV calculations. The LSPIV study reach experienced important

deposition and remobilization during the debris flow surges, therefore we did not attempt to measure the discharge rates.

**3.4 Visual velocity reconstruction**

Even though Cam 1 and Cam 3 could not be used for LSPIV, they were useful for tracking the surge fronts passing over

check dams and boulders for approximately 250 m. Some of these velocities directly covered the reach upstream from the

trap where the stage sensors were located for useful velocity comparisons. The spatial distribution of front velocities could

also be examined given the changing of slope from the reach to the trap. Even though the stage sensors are approximately 70

163  m upstream from the LSPIV area, it still gives an approximate verification of velocities.




Figure 2: Example of (A) a SfM point cloud used as a post-event reference, (B) the undistorted camera image with the reference points, (C) the orthorectified image during the 2013 debris-flow front with the sampling grid, interrogation area (IA) and the search area (SA), and (D) the instantaneous surface velocity vectors.





## 4 Analysed events

From 2011-2015 there have been four important events (Table 2; Fig. 3). The 2011 event was complex, with the first and most important surge consisting of a hyperconcentrated flow and, only Cam 1 and Cam 3 were operational at the time (Fig. 3). Therefore, LSPIV was not performed; measurements of flow velocity were performed manually (ratio of the time interval between the passage of the front and the distance between the two radar sensors) and by means of cross-correlation between the stage recordings (Comiti et al., 2014). There were no significant events in 2012.

The 2013 event featured one important surge, very typical debris-flow formation with a boulder front and the slurry like tail. The singular surge provided a convenient detailed analysis of the front, intermediate stage (transition from front to tail), and the tail (described later).

The 2014 event had a small preliminary surge (pre-surge) and four debris flow surges passing through the study reach. It should be noted that there was a discontinuous surge that stopped just upstream of the LSPIV measurements before the first measured surge passed through the reach. The first two measured surges were large enough to distinguish the front (S1 and S2) and tail (S1 tail S2 tail) and the latter two were too small and were kept undivided (S3 and S4). There seemed to be a higher water content with longer sustained fronts (compared to 2013). The S4 was unusually fast which behaved more of a wave passing through the filled-up sediment trap of highly saturated deposit.

The 2015 event was especially interesting because of the variation of surges. High intensity rainfall covered the entire catchment triggering many different source areas. The first surge (S1) had little sediment but carried a lot of large woody debris. S2 was a slower muddier flow, however cobbles and boulders were also transported. S3 was a larger and even slower muddy flow, carrying boulders, cobbles, and large woody debris. S4 is the slowest surge and a more visco-plastic flow still carrying cobbles. S5 is similar to S4 but carried less cobbles. In between these surges the low-flow material stops, the visco-plastic material waited for the next surge to push it forward. A low steady muddy flow continued for another 30 min with smaller surges. However, the sediment trap became filled creating a saturated pool of sediment making surges difficult to pass through.







Figure 3: Views of the three cameras during the 2011, 2013, and 2014 debris flows. Cam 2 was selected for the LSPIV application due to the best positioning.





**Table 2: Results of averaged LSPIV measurements, visual feature measurements on orthorectifed images, and radar sensors (70 – 150 m upstream from the LSPIV section) for identifiable surges in 2011, 2013, 2014, 2015 (no events occurred in 2012).**

| Event | Surge | Time | LSPIV | | | Visual | | Radar Sensors (70 m and 150 m upstream from LSPIV) | |
|---|---|---|---|---|---|---|---|---|---|
| | | | velocity (m s⁻¹) | width (m) | turbulence (degrees) | sediment concentration | velocity (m s⁻¹) | velocity (m s⁻¹) | avg height (m) |
| **2011** | HF surge | 18:00 – 18:30 | -- | -- | -- | low | -- | 2.6 | 0.6 |
| **2013** | S1 Front | 17:23:10 – 17:23:26 | 4.6 | 23 | 29.5 | high | 4.4 | 5.7 | 1.9 |
| | S1 Inter. | 17:23:35 – 17:23:42 | 2.5 | 11 | 24.6 | medium | 2.4 | -- | 1.6 |
| | S1 Tail | 17:23:43 – 17:24:05 | 2.7 | 11 | 23.1 | medium | 2.6 | -- | 1.0 |
| **2014** | Pre-surge | 17:13:45 – 17:15:13 | 3.3 | 8 | 34.1 | low | 2.7 | -- | 0.4 |
| | S1* | 17:22:01 – 17:22:17 | 5.1 | 16 | 35.2 | medium | 5.6 | 5.3 | 1 |
| | S1 tail* | 17:22:20 – 17:22:49 | 4.7 | 16 | 34.5 | medium | 4.4 | 4.8 | 0.5 |
| | S2 | 17:25:43 – 17:26:04 | 3.8 | 17 | 35 | high | 3.3 | 4.1 | 0.9 |
| | S2 tail | 17:26:10 – 17:27:00 | 3.5 | 17 | 32.5 | high | 2.8 | 3.6 | 0.7 |
| | S3 | 17:29:24 – 17:29:40 | 3.8 | 14 | 31.4 | high | 4.4 | 4.8 | 0.9 |
| | S4 (wave) | 17:30:13 – 17:30:21 | 6.0 | 9 | 31.7 | low | 6.9 | 3.5 | 0.7 |
| **2015** | S1 | 17:16:52 – 17:17:15 | 4.7 | 18 | 34.3 | low | 4.9 | -- | 0.8 |
| | S2 | 17:20:05 – 17:21:02 | 3.3 | 12 | 33.9 | high | 3.0 | 3.5 | 0.8 |
| | S3 | 17:23:30 – 17:24:01 | 2.8 | 15 | 29.9 | high | 1.5 | 3.5 | 1.25 |
| | S4 | 17:24:25 – 17:25:12 | 0.6 | 14 | 19 | very high | 0.7 | -- | 0.6 |
| | S5 | 17:26:54 – 17:27:39 | 0.8 | 17 | 14.5 | very high | 1.0 | -- | 0.8 |

* the first actual debris flow surge stopped between the LSPIV and the radar, it remobilized with S1.



## 5 Results

### 5.1 Surface flow velocities

LSPIV results of the three analysed debris flows were extracted from the upstream cross-section of the LSPIV reach (Fig. 1). This makes surge comparisons more accurate because it is located in a more stable and confined location, rather than the open sediment trap that fills up during the events. Mean surge velocities ranged from 0.6 to 6.0 m/s and mean horizontal turbulence ($H_t$) from 14.5 to 35.2 degrees (Table 2; Fig. 4). The instantenous velocities for the 2013 event have smaller variations compared to the other events. The minimum recording frequency was 3 fps for 2013 rather than 2 fps for 2014 and 2015 because of the available light during the storms. The highest velocity (2014 S4 had the largest variation indicating the degrading accuracy.

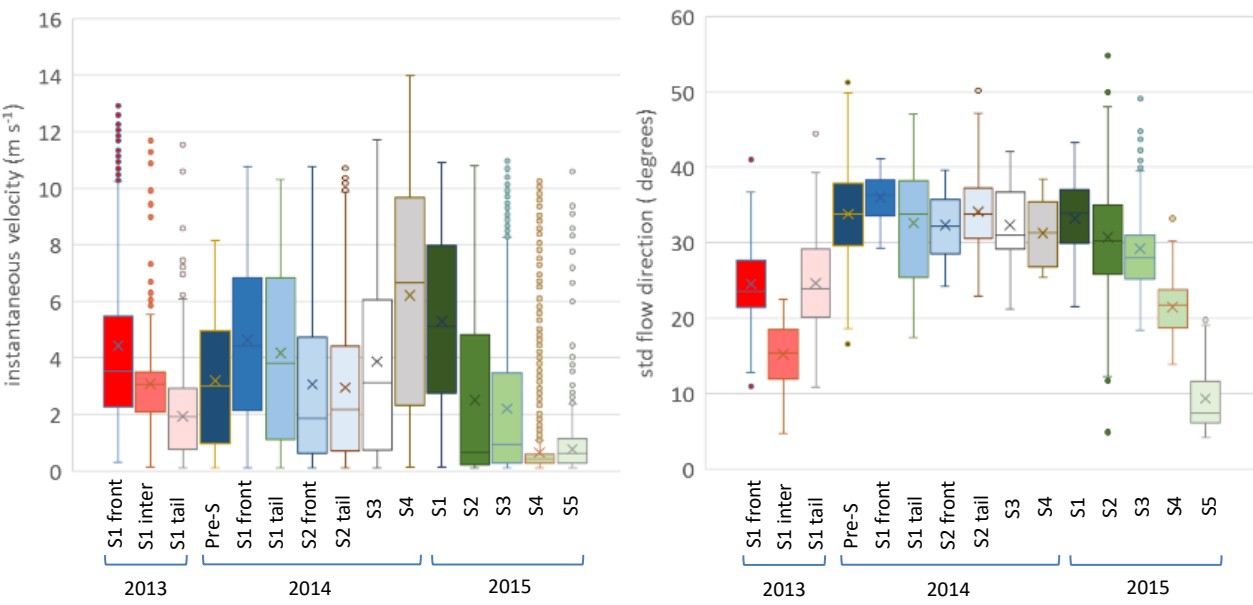

**Figure 4: LSPIV velocity and directional variation comparisons for 2013, 2014, 2015 surges located at the same cross-section.**

The LSPIV velocities seem fairly accurate considering the low camera frequency (2-3 fps), camera angle, 5cm/pixel resolution and the unsteadiness of the flows. Their average velocities at a given cross-section were compared with manual measurements of identifiable features on the same orthorectified images to validate the LSPIV cross-correlation matching (Table 2; Fig. 5). The LSPIV has a slight over estimation with a mean difference of 0.1 m/s and a standard deviation of 0.56 m/s. The LSPIV estimates are however more robust because of the large sample sizes and the feature picking does not always represent the flow velocity accurately.




The LSPIV velocites are also compared with the velocities measured from the radar sensors 70 - 150 m upstream (located in

Fig. 1).  Given the downstream decrease in velocity, the velocities are very aggreable with a mean difference of -0.45 m/s

and a standard deviation of 0.39 m/s (Table 2; Fig. 5). Not all of the surges could be traced from the radar sensors to the

LSPIV reach, rather they will stop and be pushed by the next surge. This is especially the case with the visco-plasctic surges

in 2015.

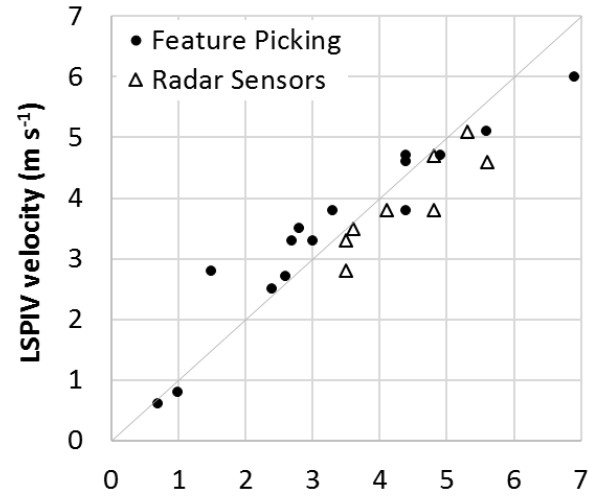

**Figure 5: LSPIV velocities compared with velocities derived from feature picking in the orthorectified image sequences and radar sensors located 70 – 150 m upstream.**

The LSPIV study was fixed within one reach which limited our perspective in the spatial distribution of the surge velocities.

Therefore we took advantage of the three monitoring cameras to generally estimate the propogation of the flow over a span

of 250 m. The visual estimates of observable debris flow fronts show a large variation of slope-velocity trends (Fig. 6). Most

of the surges are clearly dependent on slope. However, some surges (2014 S2,S3; 2015 S1, S3) have no apparent dependence

on slope because of disruptions to the flow such as log jams and very visco-plastic flows. Longer multiple reaches of LSPIV

studies will be needed to better understand the continuity of the surges and their relationship between turbulence and slope.





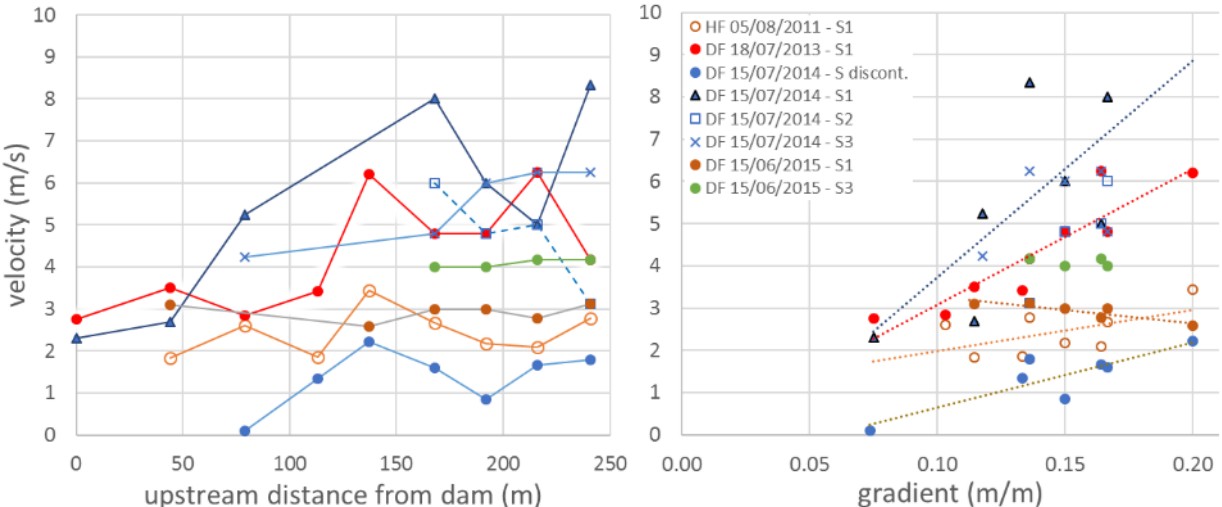

**Figure 6: Front velocities (estimated from observable fronts passing by distinct features in Cam 1, 2, and 3) are plotted according to distance from the end of the sediment trap (left) and local slope (right).**

**5.2 Pattern of flow velocities from the 2013 debris flow**

The LSPIV results can be presented and analyzed in several different ways. For the 2013 debris flow, we show the map view of the average velocities for the front, intermediate and tail (Fig. 7). Despite the classic form of the 2013 debris flow, it had a very interesting dynamic when entering the sediment trap. The front has high scattered average velocities covering the whole reach. The intermediate (transition from front to tail) shows a distinct decrease in velocity with a more homogeneous distribution. Zero velocities correspond with the boulder front deposition. The low velocity tail becomes more confined traveling around the boulder front as a more laminar flow (Fig. 8C).

Three cross-sections were examined to compare the velocity-time profiles of the event (Fig. 8). The peak velocity in the front gradually decreases in duration, nonetheless when traveling through the reach the velocity remains relatively high. For the intermediate part, there is a distinct slump in velocity where the boulder front was deposited in cross section B. The tail of the debris flow increases downstream, this is expected since the boulders confined the channel. In Figure 8, the LSPIV computation domain is overlapped on a map of the residual height, computed on the pre-event topography as the cell-by-cell difference between the SfM DEM and a smoothed mean DEM, whose cells have a value equal to the mean of the neighboring cells at a 5-m scale (Cavalli et al., 2008). The residual height (Fig. 8) shows the general form of the channel revealing the smaller confined channel along the left bank and larger convex features covering the center and right bank. These features correspond with the flow dynamics seen in Figure 8 with the boulder front depositing on the higher convex features with the water surge passing around in the lower confined channel.

The longitudinal profile of the average velocities combined with the video imagery and multi-date topography (Fig. 9) distinctly show the boulder front depositing after the sudden decrease in local slope (down to a negative slope) and channel





widening. The front average velocity remains constant even after the deposition of boulders. The intermediate part of the surge shows the correspondence of the decreased velocity and the deposition. The boulder deposit narrows the channel and therefore increases the velocity for the tail of the flow. The tail has an unusual increase of velocity at the downstream end despite the local widening of the channel with decreasing velocity. Either there was a released plugging upstream or there was important decrease of sediment concentration (upstream deposition).

Several studies observe peak velocities of debris flows located behind the boulder front (Pierson, 1986; Arattano and Marchi, 2000; Suwa, 1993). The high concentration of the interlocking boulders creates a high frictional resistance and low mobility (Pierson, 1986; Suwa, 1993). Debris flow channels typically have several reaches with important narrowing and widening and naturally the velocity longitudinal profile must adjust to each channel reach. When the front is confined, boulders interlock, velocities are higher behind the front as previous studies showed. In our case, we observe the boulders unlocking which creates more mobility where the peak velocity is in the very front of the flow. The boulders deposit as a levee because of the decrease in transport capacity.



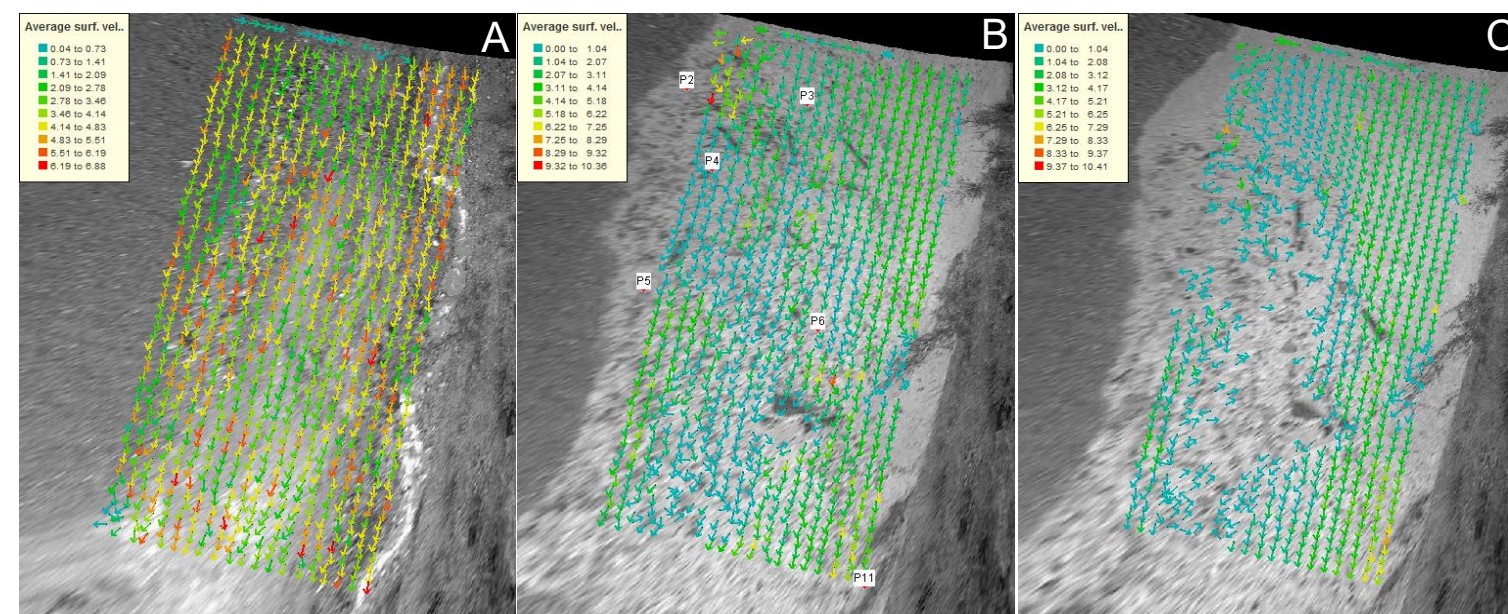

**Figure 7: Average LSPIV velocities (m s⁻¹) for the 2013 debris flow font (A) intermediate (B) and tail (C).**





**Figure 8: 2013 debris flow LSPIV velocity time series at three cross-sections (A, B, and C) with red (front), yellow (intermediate), and blue (tail).**




**Figure 9: The 2013 debris flow LSPIV average velocity of the front (red), intermediate (yellow), and tail (blue) traveling downstream (top). Local slope and the boulder front deposit (from multi-date SfM) are also plotted along the distance (center) as well as pre-event elevation and flow width of both the front and tail (bottom). At cross-section X, the boulder front is seen to deposit while the watery surge passes around it (seen on the right image) which gives constant peak velocity in the front of the surge (despite the front deposition).**

**5.3 Horizontal turbulence index**

Sediment concentration, viscosity, and yield strength are rheological parameters that can influence the turbulence and are commonly associated with flow resistance coefficients (eg. Rickenmann et al., 1999). For all the surges in 2013-2015, we found that turbulence has a strong relation with the surge velocity (Fig. 10), whereas flow heights and flow widths had little influence on the surge velocities. We compared our horizontal turbulence index ($T_h$) measurements (see section 3.3) to the




empirical flow resistance equation for debris flows from Koch et al. (1998), described in an empirical review from
Rickenmann et al. (1999):

$$C = \frac{V}{H^{0.3}S^{0.5}} ,$$ (1)

where velocity ($V$) is the average LSPIV velocity for each surge, slope ($S$) being constant, flow height ($H$) measured
upstream from the radar sensors, and the flow resistance coefficient ($C$). The $T_h$ strongly correlates with the variation of
mean surge velocities (Fig. 10). We begin to see a relationship with the coefficient by using a power-law with the velocity
function written as:

$$V = 0.005T_h{}^{2.2502}H^{0.3}S^{0.5} ,$$ (2)

however more surges need to be measured to better define the function. The influence of spatial and temporal sampling
resolutions also needs to be better understood for further application.

Laminar and turbulent flows are often defined according to Froude's critical number:

$$F = \frac{V}{\sqrt{gH}} = 1 ,$$ (3)

where below 1 is laminar and above 1 is turbulent (Enos, 1977). Froude's function ($F$) was compared with $T_h$ showing
similar trends as with $C$ (Fig. 9). Sediment concentrations from visual estimates (Table 2) were used to classify this
comparison showing good correspondence with $T_h$ versus $F$ for the debris flow fronts. Sediment concentrations for the tails
or waves did not correspond well, probably because of influences of fluid pressures from the front and the pooling of slurry
in the sediment trap. Visual estimates of sediment concentrations become difficult to classify, especially around the critical
Froude's number, however there still remains to be a strong relationship between $T_h$ and $F$ (Fig. 10) written as:

$$V = 0.0005T_h{}^{2.2823}\sqrt{gH} ,$$ (4)

For some of the surges, boulders and logs can be seen rotating, resulting in misrepresentative flow directions. Our
interrogation area (1.3 m) for LSPIV calculations was aimed to characterize the general flow characteristics where these
misrepresentations are either too detailed or have little influence on the high sampling of the LSPIV method. Higher image
resolution and camera speed might give further insight on boulder dynamics and log jamming.






**Figure 10:** **Classified by events (top), the horizontal turbulence index ($T_h$) compared to LSPIV average surge velocity (left) and flow resistance coefficient (Eq. 1) (right). Classified by visually estimated sediment concentrations from Table 2 (bottom), the $T_h$ compared to LSPIV average surge velocity (left) and Froude's number (Eq. 3) (right). The highest velocity and coefficient is an outlier influenced by impact or rolling wave since it was the last surge of the event traveling through the filled fully saturated sediment trap.**

## 6 Conclusions

We have presented LSPIV-derived velocities for three debris flow events in the Gadria channel, for a total of 11 surges and these velocities were compared with manual measurements on the ortho-rectified imagery and radar sensors. LSPIV appears to be a reliable method for measuring velocities of such type of geophysical flows, and to the best of our knowledge this is





one of the first studies on the topic. The directional variation of vectors from the LSPIV was introduced as an index of horizontal turbulence ($T_h$).

Within the studied reach, debris flows varied in velocity and turbulence between different events, between individual surges within an event, and even within each surge. Several contributing factors can explain the variation such as rainfall variability, activation of variable source areas, channel storage levels, check-dam failures, boulder and log jamming, and just the complex interactions between the channel dynamic and the flow. For example, the 2015 event distinctly had the largest variation of surge velocities and turbulence that most likely caused by the burst of rainfall distributed over most of the catchment, which in turn activated more source areas than other events. The 2013 debris flow showed that a gentle relief in the channel opening can influence the front material deposition but not decrease the mean front velocity because of the water surge passing through and around the unlocking boulders. A strong power-law relationship is found between velocity and the $T_h$ as well as the flow resistance coefficient C in the empirical equation of Koch et al. (1998). We propose that the $T_h$ measurement improves the flow resistance coefficient for estimating velocity.

Visual estimates of front velocities were also made from three monitoring cameras to quantify the spatial distribution showing various slope-velocity trends. Higher sediment concentrated and visco-plastic surges tend to stop in the channel and wait for the push of the next surge. This shows the discontinuity of the debris flow propagation that holds in question in how we can infer these observations upstream and downstream.

The LSPIV application on debris flows has shown to be very effective but there still needs to be a better understanding of the spatial and time resolution and the influence of slope. Some suggestions can be made for this type of monitoring, such as 1) be sure that the minimum frame rate of the IP camera is high enough to capture the movement ($\lesssim 2$ fps, depending on the flow velocity) or use a fixed frame rate from an analog camera; 2) locate the cameras to a stable reach with high viewing positions that are perpendicular to the flow; and 3) overlap the study area directly over stage sensors for discharge measurements for proper analysis of $T_h$. Further studies can also involve calibrating geophones with the $T_h$ which are more easily distributed in the field.

Further research on LSPIV derived velocity and turbulence needs to address the influence of confinement and roughness of the channel bed. Debris-flow channels have intermediate and large scale roughness that make flow velocities and turbulence more variable as flow heights decrease (Rickenmann and Recking, 2011; Ferguson, 2012). Large scale roughness can effect the confinement of the channel such as a large boulder or a debris-flow levee. Pre-event high resolution elevation models and their residual heights and standard deviations at varying scales (Cavalli et al., 2008) will provide better insight on spatial distrubution of debris flow velocities when they are directly compared with LSPIV measurements.





**Acknowledgements**

Funding for this research came from the research project "Kinoflow" funded by the Autonomous Province of Bozen-Bolzano. The debris flow monitoring station of Gadria catchment is managed by the Civil Protection Agency of the Autonomous Province Bozen-Bolzano. A preliminary analysis of the debris flow hydrograph conducted by V. D'Agostino and F. Bettella (Department TeSAF, University of Padova) helped interpret the 2014 event. We also thank Alexandre Hauet (EDF-DTG Grenoble) who provided guidance and advice for the Fudaa-LSPIV application.

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
