# Peer review of "Exploiting LSPIV to assess debris flow velocities in the field"

_Natural Hazards and Earth System Sciences, 2017_

## Referee Comment (RC1) · C. Graf (Referee) · 21 Aug 2017

The paper on "Exploiting LSPIV to assess debris flow velocities in the field" by Theule et al. addresses relevant scientific and technical questions and presents data derived with a novel method and a new concept. The data and method are up to international standards. All applied methods and assumptions are valid and clearly outlined. Results are sufficient to support the interpretations and conclusions and the authors reached substantial conclusions. Availability of the data, the methods used and the calculations made are well described and the results obtained are sufficiently complete and accurate to allow reproducibility by fellow scientists. The title reflects the content of the paper unambiguously and mentions the main keywords. The abstract provides a concise, complete and distinct summary of the work done and results obtained. Hence,

title and abstract are pertinent and easy to understand to a wide and diversified audience with the exception of the abbreviation "LSPIV". But this draws the interest of those who are not familiar with this technique. Mathematical formulae, symbols, abbreviations and units are correctly defined and used. The size, quality and readability of the figures are adequate to the type and data presented. The authors give proper credit to previous and related work and number and quality of the references are appropriate and accessible by fellow scientists. They indicate clearly their own contribution. Some additional references are given in the section below. The article is well structured, clear and easy to understand and there is no part of the paper that needs to be changed. The technical language is precise and understandable and the English language is of good quality, reads fluent and simple.

Details:

p.1, line 30: no hyphen in debris flow

p. 2, line 55: avoid repetition of vary. Maybe use "...different surges can vary...."

p. 15, line 272: insert "laterally" before "deposit"

p. 16, line 77: write "debris-flow front", with hyphen and -rp. 17, line 279: write "debris-flow LSPIV velocity...", include mean/average to the figure caption and add a value of the temporal rate/point density of the red, yellow and blue line

p. 18, line290: there seems to be missing something in this sentence. Please check

p. 19, line 299: reformulate the sentence. Maybe start with "We suspect, we hypothesize..."

debris flow with hyphen when used as adjective, e.g. p. 21, lines 341, 355 or p. 22, lines 359 and 360

It is not described well enough how much the effort is and how good the foundations

must be for the accuracy to be achieved.

no comments to figures or tables

Citations: Please add the new paper by Jacquemart et al. to your introduction. Maybe just add "..the application of LSPIV from video images on debris flows..." to p.2, line 60.

https://link.springer.com/article/10.1007/s11069-017-2993-1

General remarks:

Exciting, well-written paper, which reflects both the motivation, the derivation and application of the method and the initial results. Read with great interest!

---

## Referee Comment (RC2) · Anonymous Referee #2 · 23 Aug 2017

The authors present a generally well-written paper on the use of LSPIV for the monitoring and analyses of debris flows in the Gadria Creek, Italy. Given the difficulty to obtain spatially distributed measurements of debris flow velocities and rheological parameters in the field the study provides a valuable contribution in the adaption of LSPIV for such purposes. The study further analyses three debris flow events that occurred in the creek between 2013 and 2015, and considers measurement errors, debris flow dynamics, and the relationships between the turbulence and velocities of the flow. Despite these positive points I have several comments regarding the description of the method, the figures and the interpretation of the results which the authors might want to consider for further improving the manuscript. The text could also benefit from a thorough revision of the use of English language.

[Figure]

l. 59: (Stumpf et al., 2017) > 2016, Please carefully check all references.

l. 79 : "began in 2011, refer to Comiti et al. (2014)"

I feel something is missing in this sentence. Maybe ..."AND WE refer to..."

l. 81: "Two cameras are in a sediment trap..." Certainly they are not really IN the sediment trap but maybe rather ALONGSIDE?

l. 87-91: Providing specific number on the incidence angles and distances which worked/ did not work might help others to set up networks which are suitable for LSPIV. From this explanation is also seems that the network was initially not setup with LSPIV in mind. Please clarify. I'd also change ", coupled to" to "in combination with". A first look on Fig. 3 also suggests that the image quality from the two other cameras is reasonable and some further explanation would help the readers to understand why the videos could not be exploited.

Figure 1: Why is a part of the analyzed cross-section behind the camera 2? Maybe this is an error in the Figure? Even if the fish eye lens covers a wide panoramic view it will probably not cover objects which are more than 90° of the image center?

l. 101: "based from Le Boursicaud" > based ON Please carefully check the use of English language in the manuscript and consult a native English speaker if possible.

3.1 Video treatment: I feel that at some point you should provide more details about the camera system such as focal length, resolution, lens type, options for night time observations (did you test such options?), color vs greyscale, etc.

l. 107: "initially posed as a problem" > initially posed problems

l.108: "During recording of the flow events, the frequency reduced to 2-3 fps because of the low lighting of the storms. ...therefore we subsampled the images to the minimum frame rate of each flow event"

Could you please explain why exactly the frame-rate is reduced? Changes in the illumination conditions can be compensated with changes in the ISO or the aperture, rather than by the shutter speed. Does a lower shutter speed not lead to blurry images for something moving as fast as a debris flow? In addition, to avoid subsampling, you could just use the time-stamps of each frame, which can be extracted with tools such as FFMPEG. Did you check that the nominal frame rate corresponds exactly with the time-stamps of individual frames? There can be bad surprises where the actual frame rate varies even if the nominal frame-rate is set constant.

l. 111 ff.: Could you provide an estimate of the accuracy of the lens calibration? Since, you are already explaining the camera calibration; it would also seem logical to indicate here how you determined the camera orientation.

l. 123: "resolution colored point clouds (1300-2900pts/m3) making it a reliable spatial and visual reference. For the LSPIV purposes, the point clouds were rotated to make a horizontal flow plane."

The point density is not a good indicator for the quality of the surface reconstruction. Could you provide an independent measurement to estimate the accuracy of the derived point clouds? How did you georeferenced the point cloud? It is not clear why the rotation is needed and how you assured that the 'flow plane' is indeed horizontal after the rotation. The channel certainly has a significant slope, though isn't it an error to assume that it is horizontal? Please clarify.

l. 130-135: This paragraph needs further clarification. 1. What exactly is the purpose of matching the point clouds and the video imagery (determining camera orientation?) 2. You state that "Errors increase going down and across the channel according to the camera's oblique angle." Errors of what? How did you quantify them and what is there magnitude? 3. You state that "The flow plane elevation was also measured by averaging matched features touching the flow line in the post-event point cloud." So the underlying assumption is that the flow height is locally constant (i.e. a horizontal plane)? How realistic is this assumption or on other words how steep is the channel?

How does this relate to rotation to the point cloud mentioned earlier? Given that you undertake multiple measurements; could you provide an uncertainty of the plane elevations? 4. You might also want to explain briefly the rectification process in which the video frames are projected to the plane(s) since not all readers of NHESS might be sufficiently familiar with LSPIV processing.

l. 140: "(3.75 -5 m) downstream, 60 -100 pixel (3-5 m) wide, and a small 5 pixel segment upstream to capture any perpendicular flow"

What do you mean by perpendicular flow here? Please clarify.

l. 150: "horizontal turbulence index (Th). We measure Th by taking the standard deviation of the flow directions at a given cross-section for a given surge"

This is fairly simplistic indicator compared to a real decomposition in mean and turbulent flow commonly used in fluid mechanics. Its shortcomings (e.g. fluctuations in speed are ignored) should at least be acknowledged. A formula for Th should be provided since computing the standard deviation of a circular quantity requires some transformations.

3.4 Visual velocity reconstruction

The use of Cam1 and Cam3 for carrying out manual reference measurements implies that the internal and external parameters for those cameras have been calibrated. Please clarify if you used the same procedure described above for all cameras. Are the "Visual" values in Table 2 derived from Cam1 and Cam3? Given the numerous points that can be measured both automatically and manually it seems odd to provide only a single value for each section in the debris flow in Table 2. A mean and a standard deviation for both LSPIV and visual measurements would be much more meaningful.

5.1 Surface flow velocities

- "Mean surge velocities" ... Please explain earlier how the means are computed - Ht or Th? - The velocities given in Figure 4 do not fully match with what is reported

in Table 2. Assuming that the crosses mark the distribution mean (which should be indicated in the figure caption) it seems that for example that for 2013 S1 inter and S1 tail are marked at around 3.0 and 2.0 while in Table 2 values of 2.5 and 2.7 are provided respectively. Please ensure that your results are correct and consistent (also in Fig 5).
- For the interpretation of Figure 4 it would also be helpful to use the time on the X-axis and place the boxplots accordingly for each event. Presenting the cumulative rainfall pattern along with this may then also help to better clarify and interpret the events.
- ". . .the feature picking does not always represent the flow velocity accurately. . ." I agree. Feature picking is typical less precise and comprises the same systematic errors resulting from camera calibration and rectification. I might be helpful to recall this here.

Regarding Figure 6:

- Where is the dam you are referring to? Maybe indicate it in Fig 1.

- The dashed blue line (left figure) is strange and could give the impression that the flow suddenly stops after a strong acceleration. Please explain.

- Maybe indicate on the left figure which section was observed with which camera.

- The figure on the right is a bit messy. It is nearly impossible to clearly associate all trend lines with their respective points. Why do some series have trend lines while others do not?

- Given the complexity debris flows (different rheologies, kinetic energy, etc.) plus the presence of check dams (Figure 1) and changes in the channel width make it seem elusive to establish a simple relationship between velocity and slope. I would recommend adapting this section either by increasing the complexity of the interpretation or omitting the analysis.

l. 245: "of the average velocities for the front, intermediate and tail" For the interpretation it would be helpful to recall here the time intervals for each average.

Figure 7: The background images are hard to interpret for readers not familiar with the site. Maybe add some further information such as the outline of the debris flow in each image to facilitate the interpretation.

l. 250 ff: Regarding Fig. 8 I'm a bit surprised regarding the spikiness of the time-series (even after averaging along the profile). Do you consider this a plausible feature of the debris flow or could this be a residual of the LSPIV method. Figure 8: Please improve the legend and caption of the figure on the left side. What you show here seems rather a difference between two dates rather than a residual. Furthermore, please explain why the time-series is interrupted around 17:23:34.

Figure 9 also requires further improvements. Please enumerate the subfigures according to the journal guidelines and provide separate captions for each of them. Maybe indicate in Figure 8 where the profile in Figure 9 starts.

l. 305: For the sake of completeness g should be defined.

l. "We propose that the Th measurement improves the flow resistance coefficient for estimating velocity." This seems a bit curious since you measure velocity directly with LSPIV. Would it not be more useful to use Th as an alternative to estimate rheological parameters which are typically harder to derive? Please clarify.
* * *

---

## Author Comment (AC1) · 5 Oct 2017

General Remarks

Thank you for the supporting review. The minor corrections are mentioned below

Detailed Response:

–

p.1, line 30: no hyphen in debris flow

- Modified

–

p. 2, line 55: avoid repetition of vary. Maybe use "...different surges can vary...."

- Modified

–

p. 15, line 272: insert "laterally" before "deposit"

- Modified

–

p. 16, line 77: write "debris-flow front", with hyphen and -r

- Modified

–

p. 17, line 279: write "debris-flow LSPIV velocity...", include mean/average to the figure caption and add a value of the temporal rate/point density of the red, yellow and blue line

- Modified

–

p. 18, line290: there seems to be missing something in this sentence. Please check

- It seems complete, without any lengthy wording.

–

p. 19, line 299: reformulate the sentence. Maybe start with "We suspect, we hypothesize..."

- Modified to "We suspect"

–
debris flow with hyphen when used as adjective, e.g. p. 21, lines 341, 355 or p. 22, lines 359 and 360

- Modified

_

It is not described well enough how much the effort is and how good the foundations must be for the accuracy to be achieved.

- Mean differences with the radar sensors are restated in the conclusions. It is difficult to say how much effort is required to get these accuracies, we provide a field example, more study sites are needed to provide better guidelines.

_

Citations: Please add the new paper by Jacquemart et al. to your introduction. Maybe just add "..the application of LSPIV from video images on debris flows..." to p.2, line 60.

- Citation is added

---

## Author Comment (AC2) · 5 Oct 2017

General Remarks

Thank you for the detailed review. There have been key insights, particularly the turbulence analysis that have helped improve the article. Both flow direction variation and velocity variation are now used in the turbulence analysis. Improvements have been made in the methods description, figures, and interpretation of results as the reviewer has proposed.

Detailed Response:

—

l. 59: (Stumpf et al., 2017) > 2016, Please carefully check all references.

- Modified

—

l. 79 : "began in 2011, refer to Comiti et al. (2014)" I feel something is missing in this sentence. Maybe . . ."AND WE refer to. . ."

- Clarified to "for detailed information of the study site and monitoring setup, refer to Comiti et al. (2014)"

—

l. 81: "Two cameras are in a sediment trap. . ." Certainly they are not really IN the sediment trap but maybe rather ALONGSIDE?

- Modified

—

l. 87-91: Providing specific number on the incidence angles and distances which worked/ did not work might help others to set up networks which are suitable for LSPIV. From this explanation is also seems that the network was initially not setup with LSPIV in mind. Please clarify. I'd also change ", coupled to" to "in combination with". A first look on Fig. 3 also suggests that the image quality from the two other cameras is reasonable and some further explanation would help the readers to understand why the videos could not be exploited.

- This section is now clarified with incidence angle range. Distances are not so useful and can be misleading since it depends on camera resolution, and if the reader is interested, it can easily be estimated in Figure 1. It is also clarified that the other two cameras were in fact too close, limiting the area for the scale of a debris flow.

—

Figure 1: Why is a part of the analyzed cross-section behind the camera 2? Maybe this is an error in the Figure? Even if the fish eye lens covers a wide panoramic view it will probably not cover objects which are more than 90âŮę of the image center?

- The camera symbol was not appropriately angled, it is now repositioned in the figure.

—

l. 101: "based from Le Boursicaud" > based ON Please carefully check the use of English language in the manuscript and consult a native English speaker if possible.

- Grammatical overlook was corrected

—

3.1 Video treatment: I feel that at some point you should provide more details about the camera system such as focal length, resolution, lens type, options for night time observations (did you test such options?), color vs greyscale, etc.

- Camera model and resolution are added. There were no events during the night. Spotlights are triggered during rainfall in the night.

—

l. 107: "initially posed as a problem" > initially posed problems

- Modified to "initially was a problem"

—

l.108: "During recording of the flow events, the frequency reduced to 2-3 fps because of the low lighting of the storms. . . .therefore we subsampled the images to the minimum frame rate of each flow event" Could you please explain why exactly the frame-rate is reduced? Changes in the illumination conditions can be compensated with changes in the ISO or the aperture, rather than by the shutter speed. Does a lower shutter speed not lead to blurry images for something moving as fast as a debris flow? In

addition, to avoid subsampling, you could just use the time-stamps of each frame, which can be extracted with tools such as FFMPEG. Did you check that the nominal frame rate corresponds exactly with the time-stamps of individual frames? There can be bad surprises where the actual frame rate varies even if the nominal frame-rate is set constant.

- The IP camera that we used had limited features (now added to the article). No automatic ISO and aperture settings. For the subsampling, the camera copies previous frames to fill in the holes of the adjusted frame rate. We took special care that the frame rates were correct with the subsampling and validating with the time-stamps. I believe that these are confusing little details that can ruin the flow of the article.

––

l. 111 ff.: Could you provide an estimate of the accuracy of the lens calibration? Since, you are already explaining the camera calibration; it would also seem logical to indicate here how you determined the camera orientation.

- The camera orientation is not required for orthorectifying images, just an undistorted image with enough reference points (done manually in Hugin). The accuracy of the undistortion was not recorded, it is not very relevant since the images will be rectified with reference points afterwards.

––

l. 123: "resolution colored point clouds (1300-2900pts/m3) making it a reliable spatial and visual reference. For the LSPIV purposes, the point clouds were rotated to make a horizontal flow plane." The point density is not a good indicator for the quality of the surface reconstruction. Could you provide an independent measurement to estimate the accuracy of the derived point clouds? How did you georeferenced the point cloud? It is not clear why the rotation is needed and how you assured that the 'flow plane' is indeed horizontal after the rotation. The channel certainly has a significant slope,

though isn't it an error to assume that it is horizontal? Please clarify.

- The ultra-high resolution of colored points is relevant for visually matching the reference points to the images. However, yes, it should include an error, the alignment error from ICP calculations are included. The 5.7 degree rotation (approximate channel slope) is indicated. Some clarification is added mentioning it is to reduce any added spatial error.

—

l. 130-135: This paragraph needs further clarification. 1. What exactly is the purpose of matching the point clouds and the video imagery (determining camera orientation?) 2. You state that "Errors increase going down and across the channel according to the camera's oblique angle." Errors of what? How did you quantify them and what is there magnitude? 3. You state that "The flow plane elevation was also measured by averaging matched features touching the flow line in the post-event point cloud." So the underlying assumption is that the flow height is locally constant (i.e. a horizontal plane)? How realistic is this assumption or on other words how steep is the channel? How does this relate to rotation to the point cloud mentioned earlier? Given that you undertake multiple measurements; could you provide an uncertainty of the plane elevations? 4. You might also want to explain briefly the rectification process in which the video frames are projected to the plane(s) since not all readers of NHESS might be sufficiently familiar with LSPIV processing.

- 1. To clarify further the paragraph, it now begins with "For orthorectifying video images. . ."

- 2. It is the alignment error for orthorectification. Errors are indicated in Table 1. This is now clarified in the text.

- 3. Variable rough flow height is added. . . the flow height is the best estimate. There are not enough measured points to make an uncertainty (approximately 5-10 measurements).

- 4. The rectification is standard georeferencing methods such as in GIS. LSPIV process is described in the next paragraphs.

——

l. 140: "(3.75 -5 m) downstream, 60 -100 pixel (3-5 m) wide, and a small 5 pixel segment upstream to capture any perpendicular flow" What do you mean by perpendicular flow here? Please clarify.

- Changed to ". . .capture flow directions toward the banks."

l. 150: "horizontal turbulence index (Th). We measure Th by taking the standard deviation of the flow directions at a given cross-section for a given surge" This is fairly simplistic indicator compared to a real decomposition in mean and turbulent flow commonly used in fluid mechanics. Its shortcomings (e.g. fluctuations in speed are ignored) should at least be acknowledged. A formula for Th should be provided since computing the standard deviation of a circular quantity requires some transformations.

- It is a simplistic indicator, it is easier for the reader to read the explanation rather than the equation, further clarified "Standard deviation of vector orientations in 3 adjacent cross-sections for three time-steps." Fluctuation in speed analysis (Tv) is added in the methods and analysis, with the same turbulence calculation (just with vector velocity rather than direction). Th is now Td

——

3.4 Visual velocity reconstruction The use of Cam1 and Cam3 for carrying out manual reference measurements implies that the internal and external parameters for those cameras have been calibrated. Please clarify if you used the same procedure described above for all cameras. Are the "Visual" values in Table 2 derived from Cam1 and Cam3? Given the numerous points that can be measured both automatically and manually it seems odd to provide only a single value for each section in the debris flow

in Table 2. A mean and a standard deviation for both LSPIV and visual measurements would be much more meaningful.

- There had been hesitation in omitting this section, reviewers confirm our hesitation. It is now omitted

—

5.1 Surface flow velocities

"Mean surge velocities" . . . Please explain earlier how the means are computed - Ht or Th? - The velocities given in Figure 4 do not fully match with what is reported in Table 2. Assuming that the crosses mark the distribution mean (which should be indicated in the figure caption) it seems that for example that for 2013 S1 inter and S1 tail are marked at around 3.0 and 2.0 while in Table 2 values of 2.5 and 2.7 are provided respectively. Please ensure that your results are correct and consistent (also in Fig 5). - For the interpretation of Figure 4 it would also be helpful to use the time on the X-axis and place the boxplots accordingly for each event. Presenting the cumulative rainfall pattern along with this may then also help to better clarify and interpret the events. - ". . .the feature picking does not always represent the flow velocity accurately. . ." I agree. Feature picking is typical less precise and comprises the same systematic errors resulting from camera calibration and rectification. I might be helpful to recall this here.

- The data in the table were mistakenly the mean velocities for the LSPIV area (not within the cross-section). The correlations coefficients have decreased but trend lines have changed by a little in Figure 10. However, they are still important. The mean differences with velocities from the radar have improved in Figure 5.

Furthermore, the new Tv (Turbulence from velocity variation) has a much better correlation than the Td .

Tv results with also be included in Figure 4 and Figure 10 and included in the discussion.

There are multiple source areas and the rainfall data opens a door to much more analysis with sediment dynamics in the catchment, it is kept for a future article.

For Figure 4, the surge labels are easier to read, rather than the long written time periods.

—

Regarding Figure 6: Where is the dam you are referring to? Maybe indicate it in Fig 1.

The dashed blue line (left figure) is strange and could give the impression that the flow suddenly stops after a strong acceleration. Please explain.

Maybe indicate on the left figure which section was observed with which camera.

The figure on the right is a bit messy. It is nearly impossible to clearly associate all trend lines with their respective points. Why do some series have trend lines while others do not?

Given the complexity debris flows (different rheologies, kinetic energy, etc.) plus the presence of check dams (Figure 1) and changes in the channel width make it seem elusive to establish a simple relationship between velocity and slope. I would recommend adapting this section either by increasing the complexity of the interpretation or omitting the analysis.

- There had been hesitation in omitting this section, reviewers confirm our hesitation. It is now omitted

—

l. 245: "of the average velocities for the front, intermediate and tail" For the interpretation it would be helpful to recall here the time intervals for each average.

- Time periods are now included

[Figure]

Interactive
comment

—

Figure 7: The background images are hard to interpret for readers not familiar with the site. Maybe add some further information such as the outline of the debris flow in each image to facilitate the interpretation.

- Outlines and better rectified photos are added

—

l. 250 ff: Regarding Fig. 8 I'm a bit surprised regarding the spikiness of the time-series (even after averaging along the profile). Do you consider this a plausible feature of the debris flow or could this be a residual of the LSPIV method.

- This is most likely some residual turbulence in the profile and heterogeneity of the flow.

—

Figure 8: Please improve the legend and caption of the figure on the left side. What you show here seems rather a difference between two dates rather than a residual. Furthermore, please explain why the time-series is interrupted around 17:23:34.

- The figure is improved with a better legend and captions.

- The time-interuption was initially made to make a clear distinction between the front and the rest of the flow. This was a period of channel adjustment during the boulder front deposition

—

Figure 9 also requires further improvements. Please enumerate the subfigures according to the journal guidelines and provide separate captions for each of them. Maybe indicate in Figure 8 where the profile in Figure 9 starts.

- Sub-figures are enumerated and captions added. The grid is added to Figure 9 for
easier referencing to Figure 8.

—

l. 305: For the sake of completeness g should be defined.

- It is now defined

—

l. "We propose that the Th measurement improves the flow resistance coefficient for estimating velocity." This seems a bit curious since you measure velocity directly with LSPIV. Would it not be more useful to use Th as an alternative to estimate rheological parameters which are typically harder to derive? Please clarify.

- The rheological parameters can be measured in the field without video monitoring events. It is very useful for improving these estimates for modeling purposes.

---

## Author Response (AR2)

**Final Response to Editor**

Thank you for the extra time given for the submission. Below are all the minor corrections made according to your suggestions.

**P2, L55: "GoPro recording": What is this? Maybe explain.**

GoPro "video" recording - added

**P5, L105: "LSPIV calculation on the freeware…": Maybe rephrase to "the LSPIV calculations were performed using the freeware…"**

ok

**P5, L108: "resolution 1689x1345": in DPI?**

1689x1345 "pixels"- added

**P5, L117: "this used to be necessary for the Fudaa software": Unclear.**

Added – "This used to be necessary for the Fudaa software, however the more current version can now handle jpeg and tiff colored formats."

**P6, L133: Maybe rephrase section heading to "LSPIV calculations using Fudaa" or similar.**

ok

**P6, L143: Flow velocity not shown in Fig. 2C. Please give reference to this Figure when describing IA.**

corrected

**P6, L145: "search area": Please give reference to SA in Fig. 2C.**

ok

**P9, L180: "four surges": It would be good to denote S1 to S4 in brackets here.**

ok

**P9, L186: "because of the surges variable rheology": maybe rephrase to "because of the variable rheology of the surges".**

ok

**P14, L236: "17:23:10": please indicate once that these are time steps.**

Added: "time of occurrence"

**P19, L287: "strong correlation with" or "strong relation to".**

"strong relation to"

**Figure 1: This may be rearranged to fit a rectangle. Position of the study site in A should be enlarged or/and labelled in the Figure.**

There is no easy way to fit the figure as a rectangle, the position is enlarged.

**Figure 2: All images have no orientations and scales. IA and SA in C should be plotted with thicker lines, the reference to the colors in the legend should be deleted.**

ok

**Figure 3: Images have no orientations and scales. Even though these can be estimated when looking at Fig. 1, it may be convenient to indicate at least the orientations of the cameras in this Figure.**

Orientations are added

**Figure 6: Images have no scales.**

Scales are added

**Figure 8: B has no scale. In A, please reconsider the y-axes labelling ("m/m", "depth"). In the caption, it reads "along the long profile of the grid in Fig. 7". Please specify what is meant here.**

Scale is added. "Depth (m)" is now just "m".

Changed to "traveling down the long profile of the grid in Figure 7"

**Figure 9: Please denote A to D and explain the four graphs in the caption separately.**

ok